# Prevalence and Management of Phytopathogenic Seed-Borne Fungi of Maize

Rehema Erasto * , Newton Kilasi and Richard Raphael Madege

Department of Crop Science and Horticulture, Sokoine University of Agriculture,
Morogoro P.O. Box 3005, Tanzania
* Correspondence: rehemaerasto24@gmail.com

**Abstract:** Seed-borne fungi are solemn and deleterious pathogens capable of causing significant losses of quantity and quality losses in maize seeds and seedlings. They infect the crop at all points of the production chain from farms to stores. A yield loss of up to 50% can be encountered. Currently, chemical control of the disease is being implemented, though it is accompanied by several negative effects. This study aimed at identifying seed-borne fungi of maize and effective management options. A deep-freezing blotter method and morphological identification of the fungal species were implemented. The seed-borne fungi detected were *Fusarium verticillioides*, *Aspergillus flavus*, *Aspergillus niger*, *Penicillium* spp., *Rhizopus* spp., and *Curvularia* spp. However, in farmer-saved seeds, fungal incidences were significantly higher ($p < 0.01$) than in certified seeds. To identify more effective management options, the efficacy of water and ethanol-extracted bio-fungicides from three plant species, namely, neem (*Azadirachta indica*), ginger (*Zingiber officinale*), and coffee (*Coffea arabica*) were evaluated. From in vitro assays, ethanol-extracted bio-fungicides have a 100% inhibitory effect on fungal growth, whilst the inhibitory effects of water-extracted bio-fungicides are 55.88% (*Azadirachta indica*) and 46.31% (*Zingiber officinale*), followed by 5.15% (*Coffea arabica*). For the case of an in vivo assay, maize seeds treated with water-extracted bio-fungicides have higher seed germination and seedling vigor percentages. For germination, seeds treated with water-extracted bio-fungicides have higher percentages (neem and ginger (90%) followed by coffee (72.5%)) than ethanol-extracted bio-fungicides (neem (0%), ginger (2.5%), and coffee (0%)). A similar observation is made for seedling weight. Therefore, the tested water-extracted bio-fungicides can be used in treating seeds before sowing them. Further studies on effective methods of extracting bioactive compounds, and improving their shelf life, are recommended.

**Keywords:** anti-fungal; bio-fungicide; farmer-saved seed; seed-borne fungi; seed treatment



## 1. Introduction

A seed is a biological entity from which a plant's life is perpetuated [1,2]. Both formal and informal systems are used in seed production. The formal system is the one entrusted to give high-quality certified seeds [1,3]. Despite the advantages of using certified seeds, most smallholder farmers still use their recycled "farmer-saved" seeds from previous growing seasons [4–6]. This is due to the low cost, being readily available, farmers' preferable characteristics, and timely accessibility of those seeds [3,5,7]. However, most of these seeds are reported to be of poor physiological quality and are usually contaminated with seed-borne fungi including *Aspergillus* spp., *Fusarium* spp., *Penicillium* spp. *Bipolaris maydis,* and *Rhizopus* spp., leading to low-field emergence, reduced crop vigor, increased seedling diseases, and low productivity [8–13]. Before harvest, the pathogens can invade more than 50% of maize grains and produce mycotoxins such as fumonisins produced by *F. verticillioides* and aflatoxins produced by *A. flavus*. Poor seed germination and inhibition of root and hypocotyl elongation are among the phytotoxic effects of these mycotoxins [11,14–16]. In the end, a yield loss of up to 50% can be encountered due

to seed-borne fungi responsible for fungal diseases [17]. Seed treatment using chemical fungicides is a common practice in Tanzania, and also worldwide [18,19]. The continuous use of these chemicals is characterized by non-biodegradability, rapid pathogen resistance development, increased production cost, residual toxicity causing health hazards, and environmental pollution [18,20,21]. The use of bio-fungicidal extracts including those from neem, ginger, and coffee is recommended as an alternative to chemical fungicides, since they are cost-effective, environmentally friendly, and non-toxic to mammals (most of them have very low or no residual effects on plants). Different extraction solvents can be used in extracting bio-actives from plant materials. For instance, studies by Amadioha and Mondal et al. [22,23] found that the leaf extracts of neem extracted using water and ethanol were effective in reducing fungal growth in vitro and their development in plants. A review by Luzi et al. [24] reported the use of maize varieties resistant to abiotic stresses, weeds, and some diseases such as maize streak viruses; however, the knowledge and information on fungal species that can be associated with low seed germination and poor seedling vigor from farmer-saved seeds in the Mvomero district (Morogoro region, Tanzania) are lacking. Since chemical fungicides are said to have negative impacts on the environment, reduce seed longevity, and increase production costs, a search for reasonably good control methods seemed important. Therefore, the study was designed to check the quality of farmer-saved seeds used by smallholder farmers. In vitro and in vivo studies were conducted to determine the efficacy of selected bio-fungicides (since they are eco-friendly but also affordable to small-scale farmers) in managing the growth of phytopathogenic seed-borne fungi and improving maize seed germination.

## 2. Materials and Methods

### 2.1. Study Area and Duration

Farmer-saved and certified seed samples were collected from smallholder farmers in the Mvomero district and the Agricultural Seed Agency (ASA), respectively. Isolation of fungal pathogens from maize seeds was conducted at the laboratories of the International Institute of Tropical Agriculture (IITA), Dar es Salaam, Tanzania. The laboratories are located at 6.756523° S and 39.234947° E.

### 2.2. Maize Seed Varieties

The maize seeds included in the study were untreated certified STAHA, SITUKA-M1, and TMV1 from ASA, and farmer-saved STAHA, SITUKA-M1, and TMV1 from farmers. They are all open-pollinated varieties with the ability to tolerate drought but also can breed true to type when recycled. Seed sampling was performed according to ISTA rules [1]. The primary samples were drawn from bags using a diagonally inserted stick trier (762 mm trier with an outside diameter of 25.4 mm and 6 slots). For all the sources of seeds, composite samples were formed by combining and mixing all the primary samples taken from the lots of respective seed varieties. Subsequently, each composite sample of seed variety was reduced to 1 kg, labeled (variety name, source, and storage facility used), packed into cloth bags, sealed, and submitted for the testing station. The submitted samples were placed in thick paper bags of uniform size and stored in a refrigerator at a temperature of 5 °C until used [25].

### Experiment 1: Detection and Identification of Seed-Borne Fungi

The non-blocking experiment with six treatments (maize samples) was laid down using a completely randomized design (CRD). The maize samples were examined for the presence of seed-borne fungi using the moist blotter method [1]. Maize seeds (400 seeds for each sample) were surface disinfected in 1% NaOCl for 1 min and rinsed 2 times in sterile distilled water (SDW), the seeds were then left to dry for 3 min before plating for incubation. The surface sterilized seeds were placed on three layers of moistened sterilized blotter papers in sterile Petri dishes. Forty Petri dishes (each with 10 seeds) with three layers of moistened sterilized blotter papers were used for each seed sample. Seeds were deep-frozen in Petri dishes at −20 °C for 24 h. Then, seeds were incubated at 25 °C under

alternating cycles of 12/12 h of light and darkness for seven days [1,12]. Each seed was examined thoroughly under a stereo microscope (×50) (Leica, CLS 100, Brandenburg, Germany) and many fungal colonies were observed on the seed surface. Isolation of fungal colonies followed by sub-culturing to obtain pure cultures was performed [26]. During each subculture, inoculating needles were sterilized by flaming to red-hot flames after each cut, to prevent cross-contamination. Pure cultures were exposed to UV light irradiation at ×350–500 nm wavelength by alternating 12 h of light and 12 h of darkness to induce sporulation [26]. Examination of produced conidia was performed under a compound microscope (×750) [25].

### 2.3. Data Collection and Processing

The number of isolated fungal genera, species, number of maize samples colonized by each species, and number of maize samples with seed-borne fungal species were recorded [1,27,28].

$$\text{Incidence (\%)} = \frac{Number\ of\ infected\ seeds}{Total\ number\ of\ seeds} \times 100 \tag{1}$$

**Experiment 2: Efficacy of Bio-Fungicides on Seed-Borne *F. verticillioides* under Aseptic Culture**

### 2.4. Preparation of Fungal Inoculum

*Fusarium verticillioides* isolate (multiplied and stored after identification) was used in testing the efficacy of selected bio-fungicides (Figure 1).

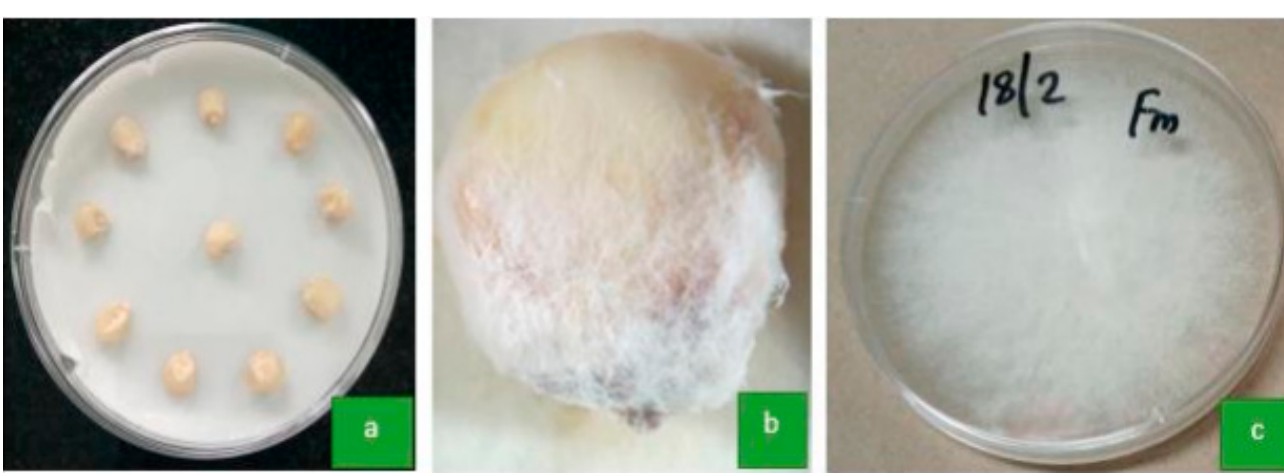

**Figure 1.** (**a**) Plated maize seeds; (**b**) Fungal growth on maize seed; (**c**) *F. verticillioides* inoculum in Petri dish.

### 2.5. Preparation of Bio-Fungicides

Liquid extracts (Figure 2) of coffee (*Coffea arabica*), ginger (*Zingiber officinale*), and neem (*Azadirachta indica*) were prepared using water and ethanol extraction solvents under aseptic conditions in the laboratory before powder of each type of plant material (50 g) was dissolved in 100 mls of SDW resulting in 50% w/v in a 500 mls conical flask [18,29–33].

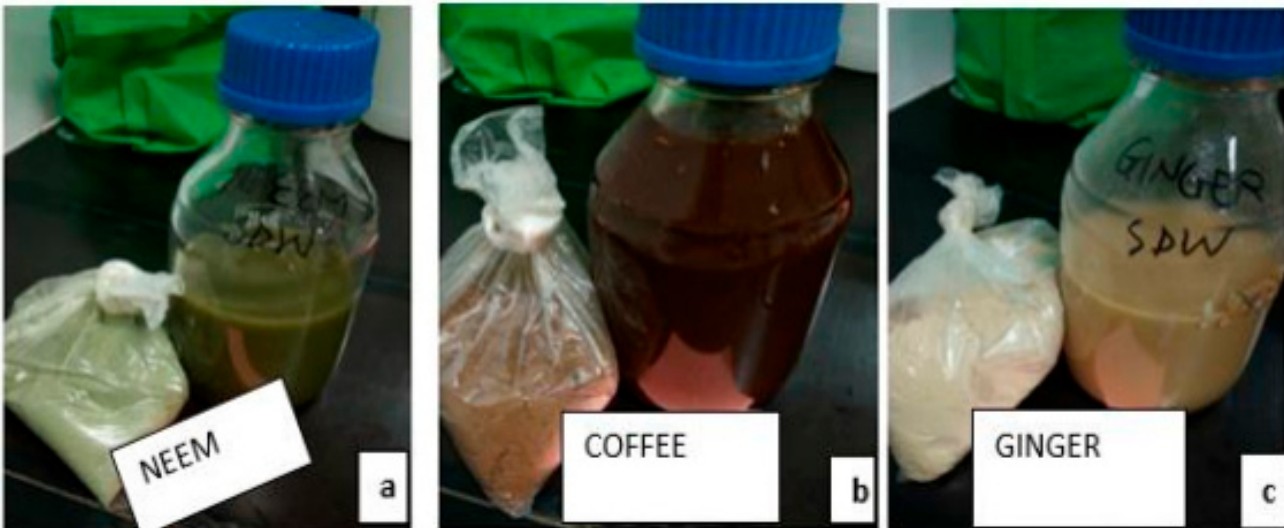

**Figure 2.** Plant extracts in powdery form (in plastic bags) and extracts dissolved in sterile distilled water (in glass bottles); the bio-fungicides. (**a**) Water-extracted neem; (**b**) water-extracted coffee; (**c**) water-extracted ginger.

*2.6. Co-Inoculation of F. verticillioides and Bio-Fungicides In Vitro*

The experiment was laid down as factorial in CRD and replicated 3 times. It involved two factors; factor A (fungicides) and factor B (extraction solvents), hence, a 5 × 2 treatment combination. PDA media was prepared [34] and the prepared bio-fungicides (50 mL of each stock solution) were added to the cool autoclaved molten PDA (150 mL) using sterile micro-filters in the laminar flow chamber. This made a 25% concentration of PDA–BF (PDA mixed with bio-fungicide) for each type of bio-fungicide. Thereafter, 20 mL of each PDA–BF was poured into a separate Petri dish and allowed to solidify [35,36]. A 2 mm inoculum disc of the 1-week-old pure culture of *F. verticillioides* was inoculated at the center of the petri dish containing the PDA–BF [37]. This followed incubation at 28 °C for 5 days and the radial growth diameter of the fungal colony was measured every day for 5 days after inoculation [18,38]. Untreated cultures served as negative control while the cultures inoculated with chemical fungicide (Apron Star® 42 WS with 20% thiamethoxam, 20% metalaxyl-M, and 2% difenoconazole) served as a positive control.

$$\% \textit{Mycelial growth inhibition} = \frac{\textit{Mycelial growth diameter (control} - \textit{treatment})}{\textit{Mycelial growth diameter control}} \times 100 \qquad (2)$$

***Experiment 3: Efficacy of Bio-Fungicides on F. verticillioides under Screenhouse Conditions***

The experiment was laid down as factorial in CRD and replicated 4 times. It involved three factors; factor A (2 seed sources), factor B (5 fungicides), and factor C (2 extraction solvents), hence, a 2 × 5 × 2 treatment combination. Four hundred seeds of certified and farmer-saved STAHA maize were inoculated by spraying $1 \times 10^5$ spores/mL *F. verticillioides* strain following procedures by Namai and Ehara [39]. The maize seeds pre-inoculated with *F. verticillioides* were later treated with bio-fungicides [18]. A total of 80 maize seeds in 4 replicates (20 seeds per replicate) of the treated seeds were planted in pots (10 seeds/pot) containing sandy soil and kept under screen house conditions. The efficacy of bio-fungicides against *F. verticillioides* was evaluated based on seed germination, seedling growth, and seedling vigor where the number of germinated emerged seedlings, dead seeds, shoot length, and weight of seedlings were evaluated 7 days after sowing (Figure 3) [1].

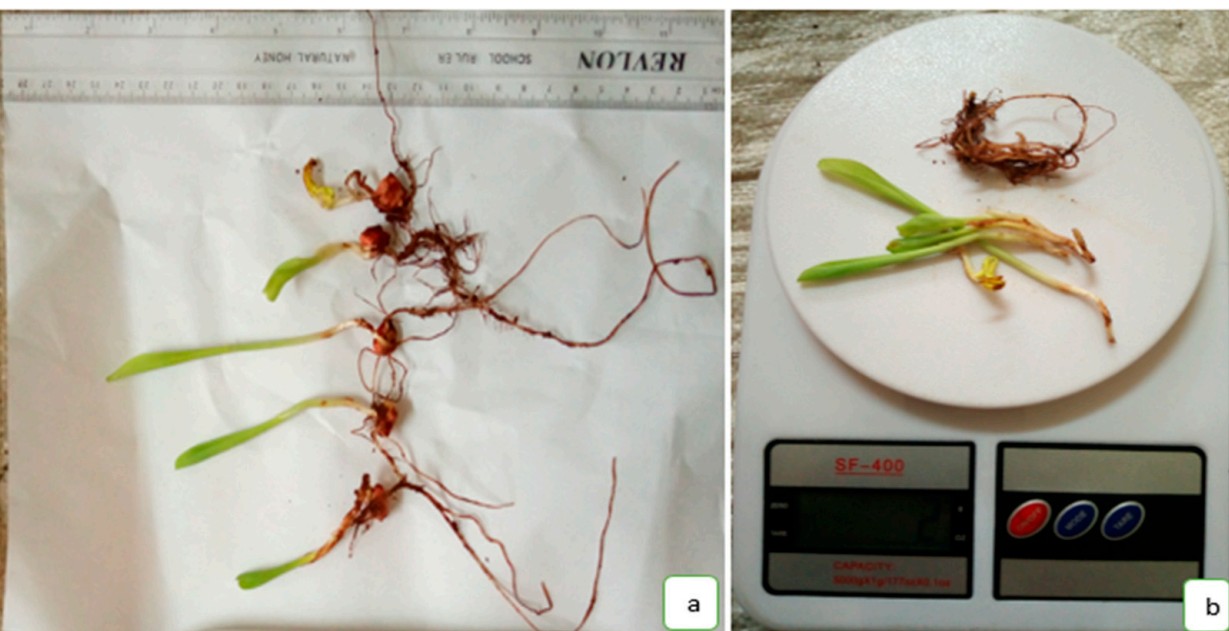

**Figure 3.** Seedlings vigor parameters; measuring (**a**) seedling height and (**b**) seedling weight.

### 2.7. Data Analysis

The Shapiro–Wilk test was performed to see if the collected data were normally distributed. The square-root and arcsine data transformations were performed (for data that were not normally distributed) before data analysis [40]. The analysis was performed based on the factorial experiment' arrangement in a CRD analysis of variance (ANOVA) model. Tukey's test ($p < 0.05$) was used in mean separation.

## 3. Results

### 3.1. Identification of Seed-Borne Fungi

Seed-borne fungal species Fusarium verticillioides, *Aspergillus flavus*, *Aspergillus niger*, *Penicillium* spp., *Rhizopus* spp., and *Curvularia* spp. were identified in all the seed samples tested. Penicillium spp. often covered/dominated one side of seeds with a profuse growth in grey–green to dark green color (Figure 4a). *Curvularia* spp. were extensively growing, covering the whole seed. They had short dark conidiophores bearing clusters of black and shiny conidia at the tips (Figure 4b). A. flavus had immature white heads, and when matured, had yellowish cream to green color heads. Also, they were observed to have long and hyaline conidiophores terminating in bulbous heads (Figure 4c). For the *Rhizopus* spp., their growth covered the whole seed and extended to the blotters (fast-spreading nature). They had brown, long, larger, and numerous solitary sporangiospores with striations. They were also more clearly visible on blotters. They changed from colorless to black with age (Figure 4d). *F. verticillioides* had whitish and characteristically powdery micro-conidia contained in chains (Figure 4e). On the other side, *A. niger* had brown to black globose conidial heads on long, erect, hyaline, and solitary conidiophores (Figure 4f). The incidences of *F. verticillioides*, *A. flavus*, *A. niger*, *Penicillium* spp., and *Rhizopus* spp. vary significantly ($p < 0.01$) between seed varieties/types and seed sources. Farmer-saved seeds have higher incidences of seed-borne fungi than certified seeds. On the other hand, the incidences of *F. verticillioides*, *A. flavus*, and *A. niger* are higher than the incidences of the rest of the identified fungal species (Figure 5).

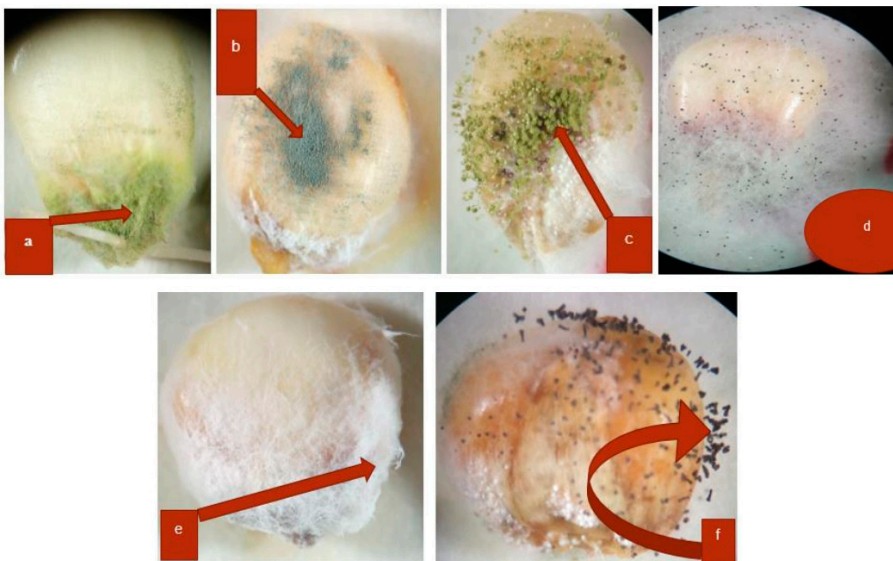

**Figure 4.** (**a**) *Penicillium* spp.; (**b**) *Curvularia* spp; (**c**) *Aspergillus flavus*; (**d**) *Rhizopus* spp.; (**e**) *Fusarium verticillioides*; (**f**) *Aspergillus niger* (both; ×50).

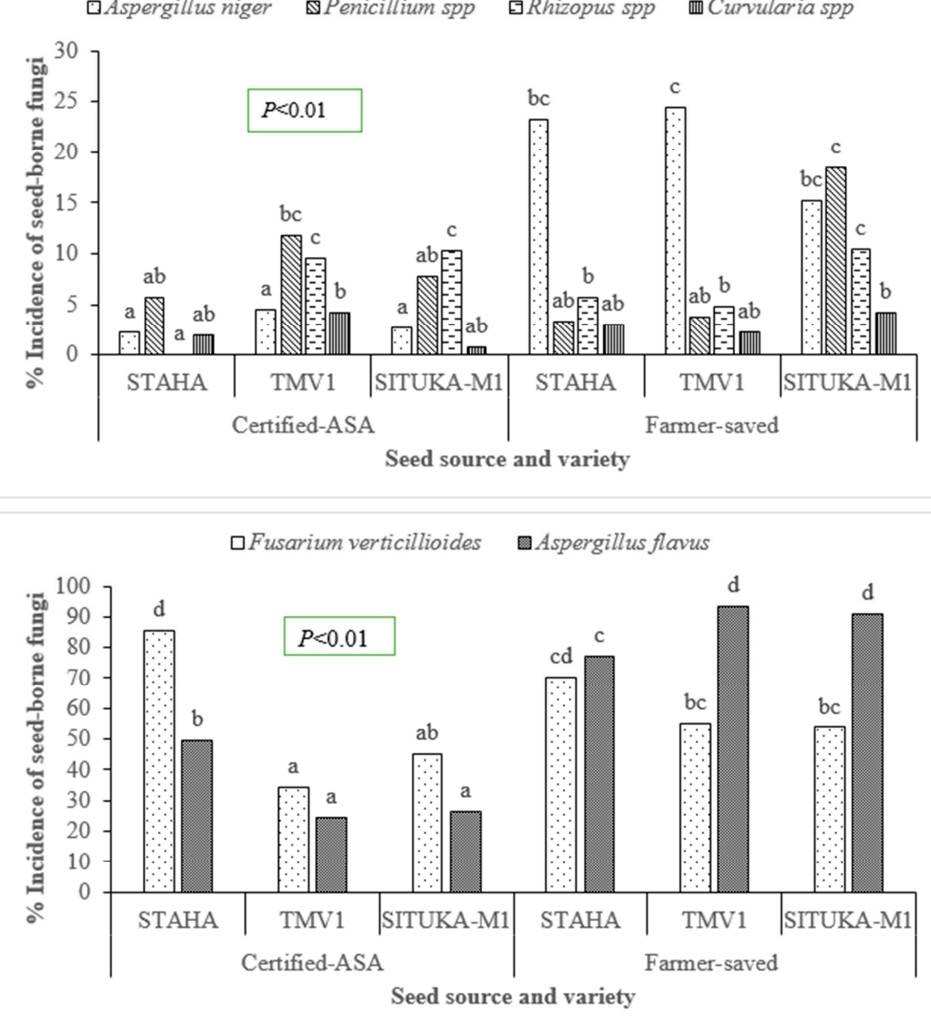

**Figure 5.** Fungal incidences. Means with the same letters along the same bars are not significantly different (*p* < 0.05).

### 3.2. Inhibition of Fungal (F. Verticillioides) Mycelial Growth

Fungal mycelial growth is 100% inhibited by all ethanol-extracted bio-fungicides while the inhibitory effects of water-extracted bio-fungicides vary with the type of plant extract. For the water-extracted bio fungicides, the highest inhibition is achieved with neem (55.88%), followed by ginger (46.31%), and lastly coffee (5.15%). The positive control, Apron Star® 42 WS, has 100% mycelial growth inhibition (Figure 6).

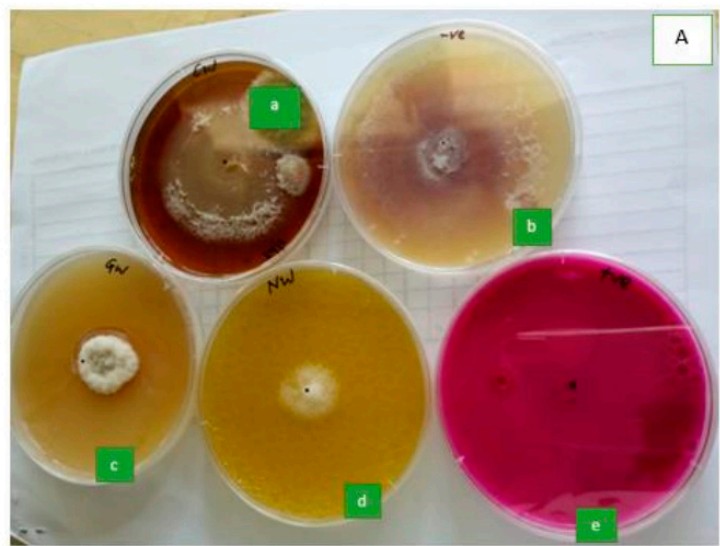

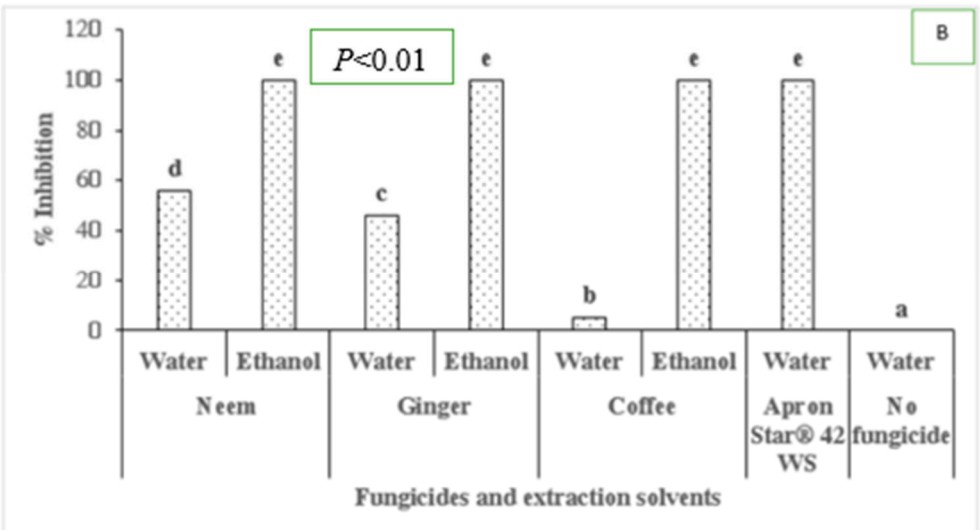

**Figure 6.** (**A**) Inhibition of mycelial fungal growth due to the application of water-extracted bio-fungicides: (a) coffee; (b) negative control; (c) ginger; (d) neem; (e) positive control (Apron Star® 42 WS). (**B**) Effects of the interaction between fungicides and extraction solvents on inhibition of mycelial growth of *F. verticillioides*. Means with the same letters along the same bars are not significantly different ($p < 0.05$).

### 3.3. Seed Germination and Seedling Vigor after Treatment

There are significant effects ($p < 0.001$) on germination and dead seeds due to interactions between seed types, fungicides, and extraction solvents. Both farmer-saved and certified untreated seeds treated with water-extracted bio-fungicides have higher percentages of emerged seedlings and seedling weight than those treated with ethanol-extracted bio-fungicides. The reverse is true for the percentages of dead seeds (Figure 7). Results in Figure 8 show that there are significant effects ($p < 0.001$) of interaction among seed types, fungicides, and extraction solvents on the proportion of normal and abnormal seedlings. From Table 1, seedling weight varies significantly ($p = 0.025$) due to the interaction between

seed types, fungicides, and extraction solvents, while seedling length has an insignificant difference ($p = 0.641$). Interaction of a solvent with the other two factors is highly pronounced because all seeds that are treated with all bio-fungicides have 0% germination. This is not the case with the seeds treated with water-extracted bio-fungicides.

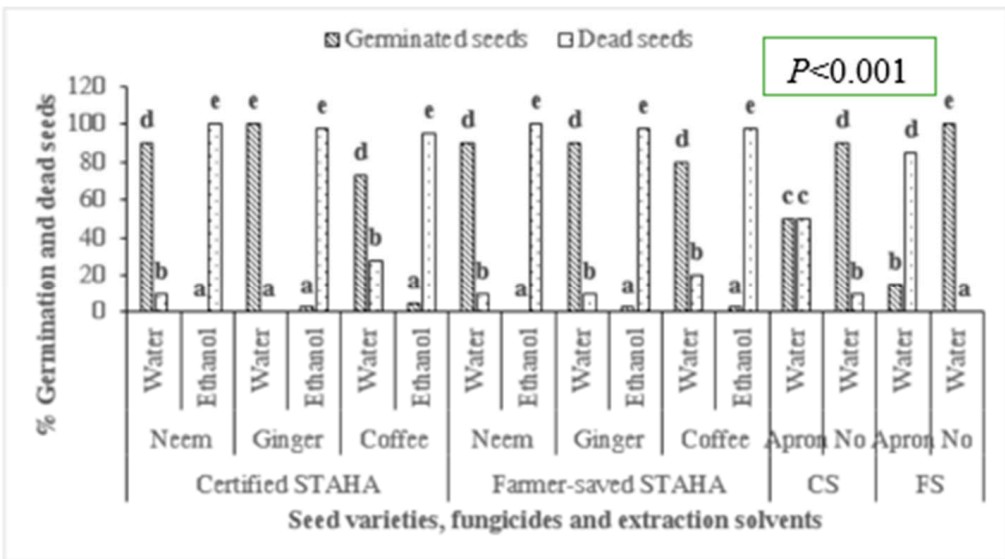

**Figure 7.** Effects of a three-way interaction between seed sources, fungicides, and extraction solvents on percentages of germination and dead seeds. Means with the same letters along the same bars are not significantly different ($p < 0.05$).

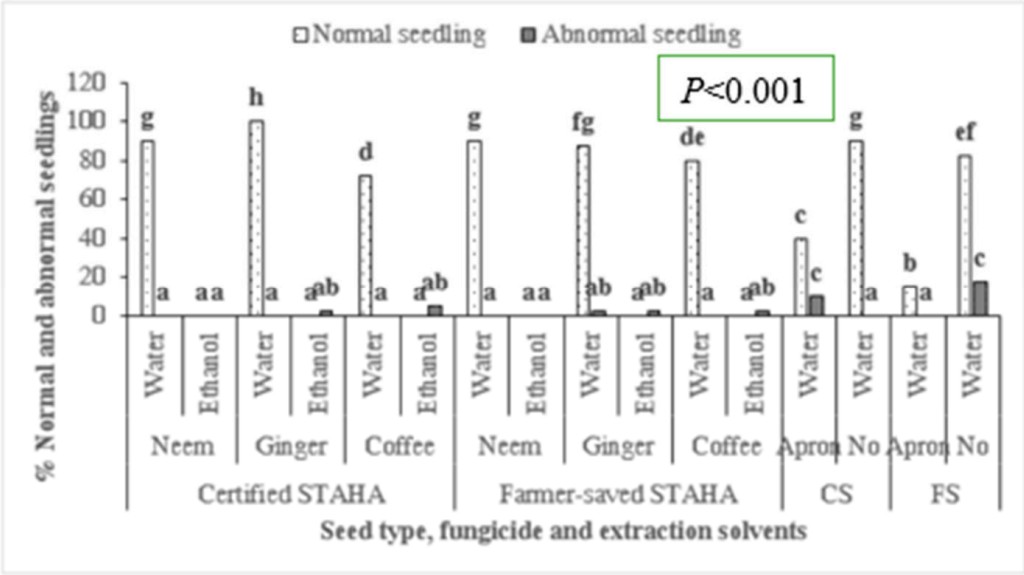

**Figure 8.** Effects of a three-way interaction between seed types, fungicides, and extraction solvents on percentages of normal and abnormal seedlings. Means with the same letters along the same bars are not significantly different ($p < 0.05$).

**Table 1.** Effects of a three-way interaction between seed types, fungicides, and extraction solvents on shoot length and shoot weight.

| Seed Types × Fungicides × Extraction Solvents (A × B × C) | Shoot Length (cm) | Shoot Weight (g) |
|---|---|---|
| Certified STAHA × neem × water | 17.1 cd | 8 ef |
| Certified STAHA × neem × ethanol | 0 a | 0.00 a |
| Certified STAHA × ginger × water | 18.38 cd | 10 ef |
| Certified STAHA × ginger × ethanol | 1.5 a | 0.2 a |
| Certified STAHA × coffee × water | 14.28 cd | 4 d |
| Certified STAHA × coffee × ethanol | 3.6 ab | 0.425 ab |
| Farmer-saved STAHA × neem × water | 20.2 cd | 10 ef |
| Farmer-saved STAHA × neem × ethanol | 0 a | 0.00 a |
| Farmer-saved STAHA × ginger × water | 21.2 d | 11 f |
| Farmer-saved STAHA × ginger × ethanol | 2.75 a | 0.325 ab |
| Farmer-saved STAHA × coffee × water | 19.1 cd | 7 e |
| Farmer-saved STAHA × coffee × ethanol | 1.28 a | 0.125 a |
| Certified STAHA × Apron Star® 42 WS | 7.3 bc | 1.5 c |
| Certified STAHA × water | 19.15 cd | 10 ef |
| Farmer-saved STAHA × Apron Star® 42 WS | 9 bcd | 0.85 bc |
| Farmer-saved STAHA × water | 17.65 cd | 9 ef |
| F-value | 0.641 | 0.025 |
| Sum of squares | 0.894 | 0.635 |
| Mean sum of squares | 0.298 | 0.212 |
| CV | 2.1 | 1.3 |

Means with the same letters along the same column are not significantly different ($p < 0.05$).

## 4. Discussion

Higher fungal incidences endanger seed germination, since they contribute to losses in the quality and quantity of seeds. The higher the incidence of fungal species on seeds the lower the germination of those seeds. This is proven by farmer-saved seeds, which have higher incidences of fungal species than certified seeds, hence, the respective effects on seed germination. This agrees with studies by Niaz and Dawar [15], which reported seed-borne fungi to be responsible for reducing seed quality and quantity, hence, poor seed germination, infecting seedlings to cause root rot, reducing seedling vigor by weakening the plant at its initial growth, and causing field epidemics. Planting seeds that are infected by mycotoxin-producing seed-borne fungi contributes to the inoculum of mycotoxigenic fungi in the soil, which later contaminates the grain with various mycotoxins [20]. Apart from primary fungal species' inoculum present in or on seeds during planting, premature harvesting and subsequent poor storage conditions of seeds are major factors contributing to a building-up of fungal species in or on seeds. This agrees with the study by Quezada et al. [41], which concludes that prematurely harvested and poorly stored seeds usually shrivel and succumb to easy attack by fungi, hence, reducing germination capacity. It is also in agreement with the notion of [42] that seeds with high rates of fungal infection have very low germination rates, which can be as low as 28% of the original potential. Moreover, this disqualifies farmer-saved seeds from having practical planting value [1].

Plant extracts with anti-fungal effects are said to be a promising solution against seed-borne fungi affecting seed germination. This is evidenced by water-extracted bio-fungicides, which, regardless of the percentage inhibition of each plant extract, all inhibit fungal growth. The inhibitory efficiency is in the order of *A. indica* > *Z. officinale* > *C. arabica*. This may be due to differences in the number of bioactive compounds found in these bio-fungicide plants. This is consistent with studies by [43–46], which report neem and ginger to have many more bioactive compounds, such as alkaloids, flavonoids, saponins, tannins, phenols, terpenoids, glycoside, anthraquinones, and steroids, and 6-gingerol, flavonoids, and phenolic acids, respectively, than coffee, which contains alkaloids alone. Furthermore, a quick degradation in bio-fungicides is reflected by an increase in mycelial diameters with an increase in the number of days after inoculation being observed. This relates to a

study by [47], which concludes that the rapid degradation of botanicals limits their use in managing plant diseases.

Contrarily, the effect of ethanol-extracted bio-fungicides on mycelial growth is observed to be higher than that of water-extracted bio-fungicides. This may be due to differences in the concentration of bioactive compounds extracted by the two extraction solvents. For instance, most of the bioactive compounds that are potent biocides are organic in nature, and ethanol (organic solvent) has a higher ability to extract those organic bioactive compounds than water, hence, a higher fungicidal effect. This finding is in agreement with the result by [23], which concludes that ethanolic extracts of neem leaves have higher fungicidal effects on *Aspergillus* spp. and *Rhizopus* spp. than crude extracts of neem. In addition to that, the current results can be related to the inhibitory effects of ethanol (which might have remained due to incomplete removal from the bio-fungicides solution) on fungal growth. Ethanol can interfere with fungal growth activities, which ends up killing them. This conforms with the study by [48] on ethanol-induced water stress and fungal growth, which reports the ability of ethanol to reduce water availability for fungal growth. It was found that about 31% of fungal growth inhibition by ethanol at 25 °C was caused by water stress, but at temperatures lower than 25 °C, the inhibitory effect due to water stress could exceed 31%, since the other non-water stress effects of the ethanol become less severe.

On seed treatment, again, water-extracted bio-fungicides improve seed germination and seedling vigor compared to seeds treated with ethanol-extracted bio-fungicides. The improvements due to water-extracted bio-fungicides can be attributed to the antifungal effect of the extracts on fungal species. Though seed treatment using bio-fungicides might not eradicate all the fungal species in or on seeds, their efficacy is still remarkable, simply because seed germination and seedling vigor from seeds treated with bio-fungicides are improved.

The death counts and weakening of growing seedlings in treatments observed for seeds treated with ethanol-extracted bio-fungicides might be attributed to the phytotoxic effects of ethanol residues on the germinating embryos and growing seedlings. A similar situation was encountered in a study by Zida et al. [33].

## 5. Conclusions and Recommendation

The findings of the study suggest the use of certified seeds because their health status is assuredly better (having lower incidences of seed-borne pathogens) than that of farmer-saved seeds. However, when they have to be used, farmer-saved seeds need to be free from, or have fewer incidences of, seed-borne fungi. Therefore, seed treatment before sowing is crucial for all certified and farmer-saved seeds. Bio-fungicides are potential candidates to be used in seed treatment because they are eco-friendly, but also because of their efficacy observed during the course of this study. Firstly, they inhibit fungal mycelial growth under laboratory conditions, but also improve seed germination and seedling vigor by hindering fungal activities on seeds and emerged seedlings. Moreover, the method of extracting bioactive compounds from plant materials must be well chosen, because extraction solvents such as ethanol, if not completely removed from the bio-fungicide solution after the extraction process, have efficient inhibition of mycelial growth, but kill the germinating embryo and or lead to student growth in the emerged seedlings. In addition to that, organic solvents are reported to be the best extraction solvents due to their ability to unlock several bioactive compounds from the plant materials. Therefore, this study calls for further studies to explore better ways on improving the effectiveness of the bio-fungicides (effective extraction of bioactive compounds and improved shelf life of the product being the priorities) in managing seed-borne pathogens for improved seed germination and seedling vigor, which, in turn, improves crop productivity.

**Author Contributions:** Conceptualization (R.E.); methodology (R.E., R.R.M. and N.K.); software (R.E. and R.R.M.), validation (R.E., R.R.M. and N.K.); formal analysis (R.E.), investigation (R.E.), resources (R.E.), data curation (R.E.), writing—original draft preparation (R.E.), writing—review and editing (R.E., R.R.M. and N.K.), visualization (R.E.), supervision (R.R.M. and N.K.), project administration (N.K.), funding acquisition (R.E.). All authors have read and agreed to the published version of the manuscript.

**Funding:** This research was funded by Sustainable Agriculture Tanzania (SAT) (3,300,000/= TZS). And the APC (200 CHF) was funded by the Authors.

**Institutional Review Board Statement:** Not applicable.

**Informed Consent Statement:** Not applicable.

**Data Availability Statement:** No new data were created or analyzed in this study. Data sharing is not applicable to this article.

**Acknowledgments:** Our sincere appreciation to Sustainable Agriculture Tanzania (SAT) for their financial support in carrying out this research.

**Conflicts of Interest:** The authors declare no conflict of interest.

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
