# Peer review of "Prevalence and Management of Phytopathogenic Seed-Borne Fungi of Maize"

_2674-1024, doi:10.3390/seeds2010003_

Round 1

Reviewer 1 Report

the manuscript entitled "Prevalence and management of Phytopathogenic seed-borne 2 fungi of maize" by Erasto et al., presents the efficacy of biopesticides against seed-born fungi. Overall merit of the manuscript is OK. SOme changes are necessary to consider before making the final decision about the manuscript

1- in the abstract authors must add an objective statement 

2- The introduction section needs improvement. First of all, authors must update the literature. Authors must use literature for only the last five years, except where some fact is being presented

 3- In the abstract problem is missing. Authors must add some statistics about the losses by the seed born diseases and some control measures along with issues with the current control measures

4- Explain some novelty of biopesticides. Why authors is interested in it

5- Present the hypothesis, research questions, and objectives in the last paragraph of the introduction section

6- Add some details of seed sample collection. As the paper is about seed-born fungi, therefore, authors must add characteristics of varieties with reference to fungal resistance 

7- In the results discussion section, authors must discuss the presence of fungus on the seeds after the treatment and also the effect of alcohol on seed germination 

8- How authors can suggest organic solvents as the best extractors when they lower the germination? Authors are advised to revise the conclusion and make recommendations based on both i.e., inhibition of fungal growth, germination, and seedling vigor  

Reviewer 2 Report

This is a interested research but needs more improving for publishing. the follow suggestions please considered even they were inappropriate possibly.

1. abstract: much more results with data(show in results part)should be showed in abstract for better understanding your conclusion.

2. all references not listed as serial number, in disorder, and some references numbers list with wrong way as "y [23] [4] [21] [10] [41] [14]" in line 35, '[34] [5] [21] [44]' in line 39, and so on. 

3. the figures in "Materials and methods" were impressive, but in my opinion, figure 1 should showed at "results" with data analysis. figure 2 needs more details introduction. 

4.  figure 3 indicated the seedling height and weight, but i am not find the data and results about seedling height and weight in "results", did i miss something or -place??

5. figure 5, the data analysis was perfect but has no significant letters note, please insert them as the same as figure 6,7 and 8.

6. results and discussion, put together corrected or not, please confirmed.

7. the results discussion need more extensively and deeper work.

8. the conclusion needs some strong data support and guarantee.

please consider the suggestion as your research realization. 

Round 2

Reviewer 1 Report

The authors have made the required changes and the paper is now acceptable. 

Author Response

Adhered to the suggestions

Reviewer 2 Report

the whole paper was revised and improved, nearly perfected.

1. the number of references still inserted disorderly (1,2,3,4, eg), please confirmed them.

2. table 1, the significant analysis of shoot length should be done and showed in the table, am i right?

3. the whole article needs more reconfirmed and revised about the sentences and expression, for better understanding.

Author Response

Adhered accordingly
